# Stem Cell-Derived Exosomes as Therapeutic Approach for Neurodegenerative Disorders: From Biology to Biotechnology

**DOI:** 10.3390/cells9122663

**Published:** 2020-12-11

**Authors:** Rodrigo Pinheiro Araldi, Fernanda D’Amelio, Hugo Vigerelli, Thatiana Correa de Melo, Irina Kerkis

**Affiliations:** 1Genetics Laboratory, Instituto Butantan, 1500, Vital Brasil St., Sao Paulo SP 05503-900, Brazil; rodrigo.pinheiro.araldi@gmail.com (R.P.A.); fernanda.damelio@butantan.gov.br (F.D.); hugo.barros@butantan.gov.br (H.V.); thatiana.c.melo@gmail.com (T.C.d.M.); 2Programa de Pós-graduação em Endocrinologia e Metabologia, Escola Paulista de Medicina (EPM), Universidade Federal de São Pauloa (UNIFESP), Sao Paulo SP 04021-001, Brazil; 3Programa de Pós-graduação em Biologia Estrutural e Funcional, Escola Paulista de Medicina (EPM), Universidade Federal de São Paulo (UNIFESP), Sao Paulo SP 04021-001, Brazil

**Keywords:** neurodegenerative diseases, extracellular vesicles, exosomes, stem cell, cell-free therapy

## Abstract

The aging population has contributed to the rapid rise in the global incidence of neurodegenerative diseases. Despite the medical advances, there are no effective treatments for these disorders. Therefore, there is an urgent need for new treatments for these diseases. In this sense, cell therapy has been recognized as the best candidate for treating incurable diseases, such as neurodegenerative disorders. However, the therapeutic use of these cells can be limited by several factors. Thus, there has been a rediscovery that extracellular vesicles, including exosomes, can be alternatively explored in the treatment of these diseases, overcoming the limits of cell-based therapy. In this sense, this review aims to revisit all areas from biology, including biogenesis and the content of exosomes, to biotechnology, proposing the minimal information required to isolate, characterize, and study the content of these vesicles for scientific and/or clinical purposes.

## 1. Extracellular Vesicles in Pathophysiology of Neurodegenerative Diseases

Since population aging has become a worldwide phenomenon, the burden of the age-related neurodegenerative diseases is expected to increase dramatically in both developed and developing nations, making these disorders a primary threat to human health [1,2].

Neurodegenerative diseases comprise a group of incurable disorders characterized by the progressive deterioration of brain function, which significantly decreases the life quality of patients [1,3]. These diseases have an impact at professional, social, economic, and family levels of patients, and they can lead to a complete inability to carry out any type of everyday activity [3]. This because the patients suffer from motors and cognitive problems with gradual memory loss and breathing difficulties [3].

Neurodegenerative diseases, which include Alzheimer’s disease (AD), Parkinson’s disease (PD), Huntington’s disease (HD), Machado–Joseph disease, multiple sclerosis (MS), amyotrophic lateral sclerosis (ALS), frontotemporal dementia, spinocerebellar ataxias, and amyloid polyneuropathy family [1,3], have as their major histopathological feature the aggregation of abnormal protein conformation in the brain as a consequence of genetic alterations [4]. Usually, the self-aggregation products lead to proteotoxic and oxidative stress, neuroinflammation, and programmed neuronal cell death, resulting in progressive neuronal dysfunction (neurodegeneration) [4].

Although the neurodegenerative diseases derive from altered proteins that undergo an unfolding process followed by the formation of β-structures and a pathological tendency to self-aggregate in neuronal cells [5,6], they have different pathophysiological mechanisms [1,4]. Despite these singularities, cumulative evidence has shown that these abnormal proteins are spread to the different anatomical brain areas through the transmission of extracellular vesicles (EVs) [6,7,8,9,10].

According to the International Society for Extracellular Vesicles (ISEV), EV is a generic term for particles naturally released from the cell in vivo and in vitro. These EVs are delimited by a lipid bilayer and cannot replicate [11]. They may serve as a vehicle for the transfer of bioactive molecules between cells [12]. Thus, it is not surprising that EVs play a role in various aspects of neuronal communication [7].

On the one hand, the EVs enhance the neuronal viability and an increase in neuron firing rate [7,13]; on the other hand, they also remove unwanted biomolecules, such as infectious isoforms or misfolded proteins [7,14], contributing to the spread of pathogenic proteins associated with neurodegenerative diseases [15].

In prion-related diseases (transmissible spongiform encephalopathies), for instance, the β-sheet conformations of prion protein (PrPSC) can anchor on the surface of EVs. Thus, this protein propagates and transmits between both cells and organisms [7,16].

In Alzheimer’s disease (AD), a late-onset neurological disorder causing a progressive loss of memory and cognitive abilities as a result of the accumulation of Aβ peptides in plaques combined with neurofibrillary tangles of tau [6,17,18], EVs can lead to AD-linked proteins and peptides spread [6].

The Aβ peptides are derived from proteolytic processing of the amyloid precursor protein (APP), which over decades was known and associated with the generation of plaques in the brain of AD patients [6]. In 2006, aiming to identify where the APP was cleaved, Rajendran et al. [19] showed that β-secretase cleaved APP on early endosomes, which subsequently resulted in the trafficking of Aβ to multivesicular bodies (MVBs), which are associated with the biogenesis of EVs.

Later, studies showed a significant reduction of plaque formation after the administration of a drug inhibiting nSMase2, which acts as reducing EV biogenesis [6,19,20]. Altogether, these data strongly suggest the EVs are involved with the pathophysiology of AD.

In a Parkinson’s disease (PD), a disease reported in 1817 by James Parkinson [21], EVs contributed to the spread of disease [5], since aggregated α-synuclein has been detected in many fluids, including brain interstitial fluid and cerebral spinal fluid [5,13,22,23,24,25].

PD represents the most frequent neurodegenerative disease of the central nervous system (CNS), affecting nearly 1% of the population over 65 years of age [6,20]. While less than 10% of PD cases are familial, the majority of PD cases are sporadic and may depend on a combination of multiple factors, including age, gender, genetics, and environmental factors [20].

The most common symptoms of PD are bradykinesia, rigidity, tremor, and postural instability [6,20,21]. The symptoms are associated with the clinical phase of the disease [6,21]. The early symptoms of the disease, which include hyposmia, disturbed sleep, and gastrointestinal dysfunction, had led to the hypothesis that PD may start in either the enteric nervous system or olfactory bulbs before spreading to other regions of the brain [6,22].

About 30% of familial and 3–5% of the sporadic PD cases are monogenic forms that derive from a single mutation in a gene that can be inherited dominantly, such as *SNCA* and *LRRK2* (also known as *PARK8*) or recessively, such as *PARK2*, *PINK1* (also known as *PARK7*) and *ATP13A2* (also known as *PARK9*) [20]. Genetic alterations in these genes are responsible for the Lewy bodies that are primarily made by α-synuclein (α-syn) [4,5,6,20]. Lewy bodies lead to the destruction of dopaminergic neurons in the substantia nigra [4,5,6].

In this sense, current studies have shown that the EVs secreted by both activated microglia and neurons play an important role in α-syn spreading and the increase of neuroinflammation, thus exacerbating neuronal dysfunction and PD progression [23,24,25,26].

The EVs are also related to amyotrophic lateral sclerosis (ALS) pathophysiology [6,8,9,27,28]. ALS is a progressive neurodegenerative disease characterized by the death of alpha motor neurons caused by heterogeneous pathogenic pathways that involve the oxidative stress, neuronal inflammation with immune cells infiltrating the CNS, mitochondrial dysfunction, errors in RNA splicing, and protein conformation [20]. ALS affects both upper and lower motor neurons and clinically is associated with weakness, muscle atrophy, fasciculation, and spasticity [4].

ALS can be subdivided into sporadic ALS (that occurs in 90–95% of cases) and familial ALS (that occurs in 5–10%) [20].

The main cause of the onset of ALS is due to the mutations in Cu/Zn superoxide dismutase gene (*SOD1*), mutations in the fused in sarcoma gene (FUS)—which encodes a protein responsible for DNA repair and related to juvenile-onset forms of the disease—or TAR DNA-binding protein 43 gene (TDP-43)—a key protein for the repair pathways of DNA double-strand breaks in motor neurons and oligodendrocytes [6,20,29,30,31,32]. However, most ALS is associated with TDP-43 mutation [4].

The TDP-43 is a 43-kDa protein that acts as a transcriptional repressor, modulating gene splicing, RNA metabolism, and stress granules [4]. In 2006, TPD-43 was discovered to be the major component of neuronal inclusions in ALS and frontotemporal lobar degeneration with ubiquitin inclusions (FTLD-TDP) [4,33].

TDP-43 is normally a nuclear protein, but in neurodegeneration diseases, it forms inclusion bodies in the cytoplasm and nucleus [4]. In ALS, surviving neurons often have TDP-43 inclusions, which can sometimes be seen in routine histologic sections but are best viewed with TDP-43 immunohistochemistry [4]. Abnormal TDP-43 can be also detected in 25–50% of AD cases, mainly in a limbic distribution [4,34,35]. As in other neurodegenerative diseases, misfolded proteins have been found in exosomes, suggesting a potential role of these vesicles in the intercellular transfer of misfolded SOD1 and TPD43 [6,8,9,27].

Already for the familial ALS, the most common cause of the disease is the expansion of hexanucleotide repeat (CGC GCC) in the non-coding region of the C9RF72 gene, which leads to a loss of protein transcription [20].

## 2. Extracellular Vesicles as Therapeutic Approach for Neurodegenerative Diseases

Despite the recent advances in the biology of neurodegenerative diseases, these disorders remains incurable, and most of the existing drugs provide only symptomatic relief and do not affect the progression of the disease [36]. For this reason, there is a pressing need to identify alternative therapies to treat these disorders [36].

On the one hand, the EVs are related to the pathophysiology of neurodegenerative diseases; on the other hand, studies have been proposed that these vesicles can be explored as a promising therapeutic tool [37]. This is because the EVs from cellular sources (bone marrow, mesenchymal stem cells, dental pulp stem cells, and others) can naturally carrier a plethora of bioactive molecules, which could reduce neuroinflammation, ameliorating the clinical signs of neurodegenerative diseases [37,38,39,40]. Moreover, these vesicles can be re-engineered to carry selected cargos, such as drugs, therapeutics, proteins, and others through the blood–brain barrier and deliver them to the brain [41,42].

Considering the emerging number of studies addressing the utility of EVs, particularly the exosomes, to treat various pathologies, including neurodegenerative diseases, this review aims to revisit the history that led to the rediscovery of these vesicles, their biogenesis, and biological properties focus on the biotechnological strategies and challenges to obtain a large production of exosomes for clinical purpose in a novel therapeutic modality known as cell-free therapy.

This section may be divided by subheadings. It should provide a concise and precise description of the experimental results, their interpretation as well as the experimental conclusions that can be drawn.

## 3. Stem Cells: The Main Source of Exosomes for Clinical Purpose

Stem cells have been proposed as the best candidate for treating incurable diseases because of their capacity to self-renew, differentiate, and produce new and healthy cells that can replace injured or diseased tissues [43,44,45,46,47].

Stem cells have been majorly classified as embryonic or somatic with the former being obtained from the inner cell mass of the blastocyte, whereas the somatic stem cells are obtained from peri-natal or post-natal sources [44].

Mesenchymal stem cells (MSCs) are the most easily accessible non-hematopoietic multipotent stem cells. These cells can be obtained from numerous adult and peri-natal tissues, such as bone marrow, umbilical cord vein, Wharton’s jelly, adipose and placental tissues, peripheral and menstrual blood, liver, spleen, and the pulp of deciduous teeth [44,45,48]. Moreover, MSCs can be propagated for several passages and show a differential potential into various cell types and lineages, including adipose, osteogenic, and chondrogenic lineages [49,50].

Over the last decade, it was well established that the MSCs have a potential role in regenerative medicine by displaying properties apart from anti-tumorigenic, anti-fibrotic, anti-apoptotic, anti-inflammatory, proangiogenic, neuroprotective, anti-bacterial, and chemoattractive effects [44,45]. Furthermore, MSCs exhibit a homing potential to the injury sites [49]. For these reasons, the MSCs have been widely explored for cell-based treatment for different diseases, including neurodegenerative disorders [36,51,52,53].

MSCs used for therapeutics may attempt the criteria defined by the International Society for Cellular Transplantation (ISCT), i.e., plastic adherence property, the expression of cell surface markers (positive for CD105, CD90, and CD73 and negative for hematopoietic markers CD43, CD45, CD14, and human leukocyte antigen marker HLA-DR) and the ability to differentiate into osteocytes, adipocytes, and chondrocytes [36].

The success of MSC transplantation relates to large-scale in vitro expansion of therapeutically qualified cells under Good Manufacturing Practice (GMP) conditions, although a standard therapeutic cell dose of MSCs, the route of administration, and the number of doses are still being optimized [44].

Furthermore, in order to avoid the risk of immunological reactions and eliminate the transmission of zoonotic disease due to the use of fetal bovine serum (FBS), MSCs have been propagated in xeno-free media [44].

Among the different types of MSCs, the bone marrow mesenchymal stem cells (BM-MSCs) are one of the thoroughly studied and well-characterized cell types in the field of stem cell-based therapy [36,54]. However, several disadvantages of using BM-MSCs have been reported. Among these disadvantages are the bone marrow isolation, which involves an excruciating surgical procedure with less yield of MSCs and lower proliferation rate, and differentiation capacity, which is correlated with the age of the donor [36]. Moreover, although the BM-MSCs can be easily expanded in vitro to obtain a large number of cells, the number of population doublings required for obtaining a sufficient number of MSCs for therapy depends on the initial number of viable cells [44]. In this sense, the few percentages of isolated MSCs from bone marrow (0.001–0.0001% only) represents the difficulty of obtaining a sufficient number of MSCs to be transplanted [54]. Thus, it is mandatory to search for an alternative source of MSCs having therapeutic benefits similar to those of BM-MSCs [36].

Moreover, to achieve a clinically significant outcome using MSC for treating central nervous system (CNS) diseases, several factors need to be taken into consideration: the dynamics of the blood–brain barrier, tissue source of MSC, and the expression pattern of homing factors such as stromal cell derived factor-1 alpha (SDF-1α), vascular cell adhesion molecule-1 (VCAM-1), hyaluronic acid, and their cognate receptors CXCR-4, very late antigen-4 (VLA-4/α4β1 integrin), and CD44, as revisited by Shamir et al. [36].

Although MSCs from several tissue sources fit the ISCT eligibility criteria, they do differ in their migratory potential and neuroprotective efficacy [36,55,56]. Thus, identifying MSCs with neurogenic potential is crucial for novel therapeutic strategies for neurodegenerative diseases.

In this sense, human dental pulp stem cells (DPSCs) emerges as a useful candidate for the treatment of neurodegenerative diseases, since these cells exhibit a high neurogenic potential [36,57].

## 4. Cell-Free Therapy: A Novel Approach to Treat the Neurodegenerative Diseases

For a long time, it was considered that the therapeutic effects of the stem cells were associated with the replacement of dead cells [45,58]. However, in a model of kidney injury, caused by the injection of toxic doses of glycerol, it was verified that transplanted stem cells remain in the injury site up to a few days and, subsequently, are not found in the tissue [58,59,60]. Similar results observed in subsequent studies [48,61,62] report that less than 1% of MSCs survive for more than one week after systemic administration [63,64].

Added to that, Takahashi et al. [65] showed that an injection of MSC conditioned medium increased the capillary density, reduced the damage size, and improved cardiac function, reducing the negative consequences of a heart attack in rats.

Altogether, these data suggest that the therapeutic effects of MSCs are mediated by their “secretome”, which is composed of a spectrum of protective bioactive molecules, which are comprised of anti-inflammatory cytokines, growth factors, neuronal factors, and antioxidants [44,66,67].

With the discovery that the secretome plays a key role in cross-talk communication between the cells and the surrounding tissues in order to mediate a biological function [43,44,45,48,68], it was postulated in a paracrine stimulation theory [58,69]. According to this theory, MSCs release a variety of bioactive molecules for intercellular communication and signaling, which are responsible for the regenerative action of these cells [54,58,70].

Up to date, several bioactive molecules were identified in the MSCs’ secretome, such as chemokines cytokines, interleukins, growth factor, lipid steroids, nucleotides, nucleic acids, ions, and metabolites [44,54]. These molecules can be found in free form or in EVs, which are recognized as a key component of paracrine secretion [71]. This attracted the interest of researchers toward the MSCs secretome that could potentially be used in cell-free therapy settings [44]. In this sense, MSC-derived exosomes can be used for treating inflammation-related diseases [71]. Furthermore, MSC-derived exosomes can be explored for use delivering chemokines, which facilitate the homing capacity of MSCs [71].

Furthermore, considering that the MSCs can be easily manufactured on a large scale, these cells are an efficient mass producer of exosomes, which can be used for therapy [45].

However, whereas the soluble biomolecules present in the extracellular medium are subjected to rapid hydrolysis and oxidative effects, the biomolecules present in EVs are more stable [58]. For this reason, cell-free therapy has focused on the EVs [7,43,72].

Cell-free therapy possesses different advantages when compared with cell-based therapy. Among these advantages are that (1) EVs can be easily prepared, stored for a relatively long period without any toxic preservative such as dimethylsulfoxide (DMSO) and transported; (2) therapeutic application of exosomes have been demonstrated to be well tolerated; (3) the use of EVs instead of whole cells avoids possible complications associated with pulmonary embolism after an intravenous infusion of MSCs; (4) avoids the risk of unlimited cell growth and tumor formation since EVs are not dividing; (5) exosomes from MSCs and epithelial cells do not induce toxicity when repeatedly injected; (6) EV may be isolated from unmodified or genetically modified human stem cells; (7) evaluation of culture medium for safety and efficacy is much simpler and analogous to conventional pharmaceutical agents [29,44,58,70,73].

Furthermore, the cell-free therapy allows the biotechnological exploration of the use of the culture medium, which is generally discarded as a by-product of the in vitro expansion of MSCs. This is because this culture medium—also termed conditioned medium (CM) [44]—is an important source of bioactive molecules, which can be found in a free form or in extracellular vesicles (EVs).

## 5. Brief History of Extracellular Vesicles Discovery

Naturally, both prokaryote and eukaryote cells produce and release different types of EVs, which can be detected in several body fluids, such as blood, semen, ependymal fluid, urine, amniotic fluid, bile, bronchoalveolar lavage fluid, malignant and pleural effusions of ascites, cerebrospinal fluid, breast milk, saliva, sweat, synovial fluid, and feces, participating in a variety of physiological process [29,48,66,72].

However, the most notable study describing the existence of EVs was published in 1946 [74]; until the 1980s, the EVs were considered as platelet “dust” or cellular debris that directly budded from the plasma membrane [66,75].

With the discovery that the EVs participate in diverse physiological and pathophysiological processes, providing intercellular communication, the studies involving these vesicles gained wide notoriety [6,58].

Depending on their size, biogenesis mechanisms, or function, the EVs are usually classified as microvesicles or ectosomes, exosomes (30–150 nm), or apoptotic bodies (generally > 1000 nm) [7,43,72].

Microvesicles (100–1000 nm) are budded as small membrane protrusions around a small portion of cytoplasm, which is released from the cell membrane surface to extracellular milieu through calpain activation, calcium influx, and cytoskeleton reorganization [76]. In contrast, exosomes are smaller (30–100 nm) and originated from endosomal vesicles through secretion from intracellular luminal space [76]. The extensive plasma membrane budding during apoptotic blebbing forms MVs and apoptotic bodies, which have much larger sizes (1–5 μm) [76].

The differential molecular expression among the EVs suggests that according to their biogenesis and size, each vesicle type exerts specific functions [6]. In this sense, while the apoptotic bodies are related to immune responses [77], microvesicles and exosomes are related to a plethora of responses, including cell signaling.

Although cumulative evidence have been shown that both microvesicles and exosomes exhibit reparative and regenerative properties, the exosomes are the most well-characterized and widely used vesicle for cell-free-based therapy [48]. In addition, they are less heterogeneous than the microvesicles [77] and do not elicit acute immune rejection nor confer risk for tumor formation [45]; thus, the exosomes have been considered a useful candidate for the treatment of several diseases, including neurodegenerative disorders [54,58,71,78,79].

One of the features that makes the exosomes useful candidate for therapy is their thermostability, once the exosomes remain stable even at a temperature range from −80 to 42 °C [80,81]. Moreover, the exosomes can be explored as vehicles for drug delivery, since these vesicles can package a plethora of molecules, including chemotherapeutics [71,82,83,84]. By exhibiting these features, the study of exosomes is an active area of research [29,71]. For this reason, this review focuses on the therapeutic use of exosomes as a novel approach to treat neurodegenerative diseases.

## 6. The Biology of Exosomes

Exosomes were described in 1983 when Harding et al. [85] firstly reported the presence of transferrin receptors in the membrane of small vesicles released from maturing blood reticulocytes into the extracellular space by a process of receptor-mediated endocytosis and recycling. However, the name “exosomes”, used as a reference to these vesicles, was introduced in 1987 by Rose Johnstone [86,87]. From the evidence collected during the 1989s that the exosomes were units that mediated the cell-to-cell communication, the term was adopted for nanosized (30–150 nm) extracellular vesicles released during reticulocyte differentiation [66,88].

Currently, it is well-established that exosomes are released from all cell types, reinforcing that their secretion plays an important role in the intercellular transfer of information [7,29,45,71,72].

Exosomes are recognized as small vesicles (traditionally with <150 nm) and with a density in sucrose of 1.13–1.19 g/mL, surrounded by a phospholipid membrane, containing abundant cholesterol, sphingomyelin, ceramide, lipid rafts, and evolutionarily conserved biomarkers, which are used to distinguish them from microvesicles or apoptotic bodies [7,45,71]. Among unique proteins found in exosomes are tetraspanins (CD9, CD63, CD81, and CD82), heat shock proteins (Hsp60, 70, and 90), major histocompatibility component (MHC) classes I and II, Alix, Tsg101, lactadherin, and lysosome-associated membrane glycoprotein 2 [43,45,89,90,91], as illustrated in Figure 1. Exosomes can also contain cell type-specific proteins, messengers [48], components of cytosol, including mitochondria, endoplasmic reticulum, and Golgi apparatus [66].

Exosomes are generated from an inward budding of endosomes, which form an intraluminal vesicle inside the endosomal compartment known as multivesicular bodies (MVBs) [29,43,72,93].

Different mechanisms for MVBs biogenesis have been postulated. In the model proposed by Trajkovic et al. [94], it was demonstrated that ceramide, generated from sphingomyelin hydrolysis by neutral sphingomyelinase 2 (nSMase2), induces negative membrane curvature via its cone-shaped structure leading to intraluminal vesicles (ILVs) budding to MVBs.

Interestingly, the microvesicle cargos are similar to the plasma cells due to simple diffusion, while the cargos of exosomes differ from the parental cells [43]. These data suggest that the cargo loading of exosomes is controlled and regulated in order to confer a selective cargo-sorting [43].

Although the selective cargo-sorting mechanisms remain not fully understood, studies have shown that the endosomal-sorting complex required for transport (ESCRT) family plays a key role in cargo loading and exosome biogenesis [7,93,95].

The ESCRT machinery is the most extensively described pathway of MVB biogenesis and ubiquitinated protein sorting into ILVs [93].

The ESCRT machinery comprises four protein complexes (ESCRT-0, -I, -II, and -III) along with accessory proteins (Alix, VPS4, and VTA-1) that sequentially act to bind future exosome cargoes and form ILVs [7].

The sorting process is initiated by ESCRT-0, which recognized and retains ubiquitinated proteins in the late endosomal membrane [93]. After initial involution of the limiting membrane into MVB lumen triggered by ESCRT-I/II, ESCRT-III forms a spiral-shaped structure that constricts the budding neck and the ATPase VPS4 drives membrane scission [7,93].

Although it was originally thought that the ESCRT machinery comprised a totally essential set of protein complexes that progressively execute stages of ILV formation, recent data have suggested that earlier components of the pathway may be more essential than later ones [7]. This is because while knockdown of the ESCRT-0 and -I proteins Hrs, STAM1, and TSG101 decreases exosome release, the knockdown of other ESCRT components can have no effect or increase exosome release from cells [7].

However, the process of protein sorting can also occur independently of ubiquitination. In 2012, Baietti et al. [96] demonstrated that the heparan sulfate proteoglycan syndecan mediates the exosome formation through ESCRT-III. This is because the interaction between syndecan and ESCRT is mediated by the small cytosolic adapter protein syntenin, which connects syndecan to the ESCRT-III-associated protein ALIX [93,96]. Trimming of the heparan sulfate chains by heparinase triggers syndecan clustering, stimulating syntenin–ALIX–ESCRT-mediated sorting and exosome production [93,96]. Heparinase also stimulates selective cargo sorting, determining CD63, but not CD81, flotillin, and CD9 incorporation into exosomes [93].

In this process, the direct interaction with tetraspanins-enriched microdomains facilitates their sorting into exosomes [7,93].

Upon maturation, MVBs fuse with the plasma membrane to secrete exosomes or degrade their cargo by fusing with lysosomes [93]. Although what distinguishes the MVBs to be secreted from those that will be degraded remains unclear, it is known that the fate of MVBs can change in response to cellular conditions [93]. For example, under starvation conditions, MVBs are degraded by fusion with autophagosomes, resulting in decreased exosome release [93,97].

The MVBs destinated for exocytosis are transported to the plasma membrane along microtubules by the molecular motor kinesin, including Rab proteins (Rab11, Rab27, and Rab35) [54,93,98]. After transport and docking to the plasma membrane, secretory MVBs couple to the soluble N-ethylmaleimide-sensitive component attachment protein receptor (SNARE) membrane fusion machinery [93].

Finally, MVBs fusing with the plasma membrane results in exosomes release into the extracellular space, where they interact with the extracellular matrix, influencing cells in the microenvironment or, enter the circulation via lymph nodes or blood (paracrine signaling) [7,93]. The secretion process results in a large number of exosomes being released in the body, with estimates of 3 × 10^6^ exosomes/μL of blood serum [6]. Alternatively, exosomes can interact with the parental cells, resulting in autocrine signaling. An alternative fate for MVBs is fusion with lysosomes, which leads to the degradation and recycling of their proteins, nucleotide, and lipid components [7].

Several steps could affect the intracellular decision to degrade or release exosomes, including the intracellular transport of MVB along microtubules to the plasma membrane, the creation of docking sites at the plasma membrane, or the recruitment of soluble N-ethylmaleimide-sensitive factor (NSF) attachment protein receptor (SNARE) proteins that mediate fusion with either lysosomes or the plasma membrane [7].

Considering that the topology of exosomes is similar to that of cells, these vesicles can interact with receptor cells, as illustrated in Figure 2 [99,100]. For this, exosome may directly interact with cognate receptors located on the plasma membrane [7]. For the delivery of RNAs or cytoplasmic proteins, exosomes may not only bind but also release their contents into recipient cells by fusion either directly with the plasma membrane or with the endosomal membrane after endocytosis [7].

In this process, several factors are mandatory to determine the likelihood of the exosome–cell fusion occurring. Firstly, it is necessary for the close apposition of the two membranes due to ligand–receptor binding or glycoprotein interactions [7]. Next, the lipid composition of the exosomes and cellular membrane are likely to affect their propensity to fuse [7,99]. In addition, acid pH in the extracellular environment enhances exosome–membrane fusion, suggesting that the acid environment found in endosomes may enhance exosome fusion after exosome uptake [7,99]. The uptake of exosomes into the endocytic system appears to be the most common mode of uptake of these vesicles [7]. In this sense, a recent study showed that exosomes may initially bind to filopodia and rapidly move inward to be internalized at endocytic hotspots at the filopodial base [101].

## 7. Exosome-Mediated Horizontal Transfers of Nucleic Acids

The exosomes are a key component of paracrine secretion, since these vesicles contribute to the horizontal genetic transfers between stem cells and tissue-injured cells [80].

The first evidence that the EVs could lead to the horizontal transfer of RNA came in 2006 from the observation that embryonic stem cell-derived microvesicles could reprogram hematopoietic progenitor cells [102]. In two independent studies, Valadi et al. [103] and Skog et al. [104] demonstrated that mRNA transported by EVs can be translated into protein, providing strong evidence of horizontal transfer of genetic material between cells.

Apart from mRNA, nowadays, it is known that exosomes are highly enriched in non-coding RNA species, such as microRNAs (miRs) and long noncoding RNA (lncRNA), reinforcing that VEs are enrolled in gene regulation [45].

miRs are small non-coding RNA molecules which are 18–22 nucleotides in length; they are produced as inactive precursors in the nucleus that undergo multiple-step processing that involves enzyme cleavage and subsequent exportation into the cytoplasm [54]. Upon functional maturity, miRs post-transcriptionally regulate gene expression to affect multiple cellular functions including cell survival, proliferation, and differentiation [54].

Interestingly, studies have been suggested that miRs are incorporated into exosomes using a regulatory mechanism that controls the RNA sorting [105,106,107]. Although the mechanisms that govern this process remain in the study, a number of defined RNA binding proteins (RBP) have been suggested as responsible for the RNA sorting [93].

In 2014, Melo et al. [108] showed that breast-cancer-derived exosomes are able to process pre-miRNA into mature miRNA, demonstrating for the first time that mature miRNAs generated within these cancer exosomes influence the transcriptome of non-malignant target cells promoting transformation [93,108]. This discovery brought strong evidence that EVs can act in the post-transcriptional regulation by paracrine form.

Five years later, in 2019, Xu et al. [109] showed that microRNA-16-5p-containing exosome derived from bone marrow-derived MSCs inhibits proliferation, migration and invasion in colorectal cancer cells by the downregulation of integrin α2 (ITGA_2_). In the same year, Che et al. [110] observed that the exosomes derived from miR-143-overexpressing MSCs inhibit cell migration and invasion in human prostate cancer by downregulating TFF3. These results showed that the miRs contained in exosomes can regulate the epithelial–mesenchymal transition (EMT), reinforcing that the EVs are able to regulate the cell (de)differentiation.

In 2018, Ma et al. [111] showed that miR-132 could be delivered by MSC-derived exosome using electroporation in order to promote angiogenesis in myocardial infarction. Using the same strategy, Chen et al. [112] showed that exosomes derived from bone marrow MSCs carrying miR-125 by transfection protect against myocardial ischemia–reperfusion injury via targeting SIRT_7_. Combined, these studies show that exosomes can be used as vehicles to deliver miRs for therapeutic purposes.

In 2014, Lee et al. [102] showed that the MSCs-derived exosomes deliver miR-124 to neural cells and induce their differentiation and glutamate transporter expression. Currently, Wei et al. [103] showed that the increased expression of hypoxia-inducible factor 1 alpha (HIF-1α) in AD decreases the miR-223. However, the MSCs-derived exosome miR-223 protects the neuronal cells from apoptosis through the phosphoinositide-3-kinase (PI3K)/Akt pathway activation [103]. In 2018, Cui et al. [104] showed that exosomes derived from hypoxia pre-conditioned MSC increase the levels of miR-21 in the brain of mice with Alzheimer’s disease (AD), preventing pathological features by regulating inflammatory responses and restoring synaptic dysfunction.

Altogether, these data suggest that the RNAs contained in exosomes play a key role in tissue regeneration.

In addition to mRNA and non-coding RNAs, studies have reported the presence of fragmented genomic DNA (gDNA) [113], including retrotransposons [114], mitochondrial (mtDNA) [105,115], and even parasitic DNA in exosome cargo [116].

In 2014, Thakur et al. [117] demonstrated for the first time that the majority of DNA associated with tumor exosomes is double-stranded. Since then, studies have been focused on the detection of double strand DNA (dsDNA) contained in exosomes as biomarkers in cancer diagnosis and monitoring [118,119]. However, the role of the DNA contained in exosomes in the regulation of target cells remains unclear.

## 8. Proteins Present in Exosomes

Despite displaying similar phenotypic characteristics, it has been reported that MSCs differ significantly in their gene expression patterns and not unexpectedly demonstrated heterogeneity in the secretome profile as well, with differences attributable to the source of the MSCs, the age of the host, and health status [44,120,121].

The abundance of cargos identified from MSC-derived exosomes has been attracted to broad attention because of their therapeutic potential [45]. However, although MSC-derived exosomes have the same morphology, they are quite different in regard to protein and RNA composition [45,122]. This because the properties of MSC-derived exosomes change according to the cell type [6].

Over 900 species of proteins were already identified in MSC-derived exosomes according to ExoCarta (http://exocarta.org). With the exception of some common proteins involved in cell metabolism and the cytoskeleton, many proteins are tissue-specific [45].

As an example, studies have identified glycolytic enzymes involved in the ATP synthesis of glycolysis and glucose transporters (GLUT) in exosomes derived from MSC [45] and cardiomyocytes [123]. However, the increased glycolytic rate leads to oxidative stress. However, it is interesting that this oxidation stress is reduced via peroxiredoxins and glutathione S-transferase in MSC-derived exosomes [124]. These data suggest that replenishing glycolytic enzymes to increase ATP production and additional proteins to reduce oxidative stress through exosomal transportation may help reduce cell death in myocardial ischemia/reperfusion injury [45].

Despite the singularities of the proteomic profiler of exosomes, now, it is well established that MSC-derived exosomes containing vascular endothelial growth factor (VEGF) and metalloproteinases (MMP-9), which play a vital role in stimulating angiogenesis, which is fundamental for tissue repair [45]. Moreover, MSC-derived exosomes also harbor cytokines and growth factors that contribute to immunoregulation [45]. In addition, exosome-specific surface proteins (CD9, CD63, and CD81) may affect the immune response by regulating cell adhesion, motility, activation, and signal transduction [45,125], reinforcing the therapeutic potential of these vesicles.

In a proteomic study, Kang et al. [126] identified 103 proteins from human neural stem cells-derived exosomes, which found an imparity between exosomes larger than the baseline (50 nm) and those smaller morphologies. These findings may explain the phenomenon recently observed by Caponnetto et al. [127] regarding the size-dependent cellular uptake of exosomes by target cells. Thus, the heterogeneity of exosomes is likely reflective of their size, content, functional impact on recipient cells, and cellular origin [29].

Considering that the secretome content is affected by a multitude of extraneous factors, the in vitro conditions for expanding the MSCs can also affect the exosome size, content, and secretion [54]. In this sense, it is known that different types of stress, such as hypoxia, irradiation, injury, and cellular stress, can be biotechnologically manipulated in order to stimulate exosome production for specific therapeutic purposes [48,128,129].

In this sense, Bartaula-Brevik [130] reported that MSC subjected to hypoxia were richer in pro-inflammatory and pro-angiogenic cytokine expression as compared to their counterparts cultured under normoxia. Moreover, the overexpression of HIF-1α in MSCs increases the exosome secretion, as well as alters the miR payload [131].

Furthermore, it is known that the starvation of fetal serum for at least 12 h stimulates the EVs secretion [132,133]. Moreover, the starvation of fetal serum is crucial for proteomic studies of the exosome [66,133]. This is because considering that some components are secreted at lower concentrations of nanograms to picograms, the serum or any growth supplements in the culture medium can overlap with and interfere with the detection and analysis of proteins secreted by cultured MSCs [44].

## 9. Strategies to Isolate Exosomes

Although several strategies have been successfully used to isolate exosomes, they represent the main obstacle to the therapeutic application of EV, since these procedures are time-consuming and generally provide few quantities of EVs [58]. However, novel methodologies have been proposed to solve these problems. Considering that the method used to isolate the EVs is critical to guarantee the physical integrity of vesicle cargo [66], this review also aims to discuss the pros and cons of each isolation technique.

Several strategies have been proposed for exosome isolation, as summarized in Table 1. However, there is not a universally accepted approach [48].

Among these strategies, the ultracentrifugation (UC) and commercial kit rooted in polymer-based precipitation are the most well established and common methods used for isolating exosomes [45], being adopted as a strategy in about 81% of research studies [72].

Ultracentrifugation-based methods can be divided into two major types of techniques according to the separation mechanism: differential ultracentrifugation and density gradient ultracentrifugation [72]. For both techniques, death cells, cellular debris, and large EVs (>200 nm) are separated using low centrifugal forces (300–2000× *g*) for 10–30 min at room temperature, as verified in the most protocol (Table 1) [72]. An additional filtration step using a 0.22–0.45-μm membrane filter can be used to increase the exosome purity.

In differential ultracentrifugation, different particles are separated using a serial of differential centrifugal forces (100,000–120,000× *g*) and time (70 min to 12 h) [72]. The pellet of exosomes is washed with phosphate saline buffer (PBS) or 0.9% NaCl solution to remove remaining proteins co-isolated with the EVs.

Differential ultracentrifugation provides pure EVs for both scientific and clinical purposes. However, the majorities of UC-based proposed methods are laborious, time-consuming, and are not suitable for the mass-scale production of EVs, making it difficult for therapeutics.

Density gradient ultracentrifugation (DGUC) is employed a sucrose density gradient, which reduces the destructive effects of centrifugal force on exosomes [72,134]. According to the exosome buoyant density in aqueous sucrose (1.10 to 1.20 g/mL), the exosomes can be easily isolated [71,72]. Although this method provides the highest efficiency for exosome purification, the suitability of this technique for clinical purposes is questionable due to the difficulty in upscaling and automating the process [48,137]. Moreover, for this method, the wash step is mandatory to remove eventual residues of CsCl or sucrose used to obtain the gradient density.

Another strategy commonly employed to isolate exosomes, especially for commercial kits, is coprecipitation. This method uses polymers, such as polyethylene glycol (PEG) 6000 or 8000, which are able to coprecipitate with hydrophobic proteins and lipid molecules, which are present in exosome membranes [72]. Although most simple and less expensive than the methods based on ultracentrifugation, the isolation using coprecipitation is not scalable, limiting its use for therapeutic purposes. Moreover, this technique requires the sample incubation with the polymers for a long time (generally 12–16 h). Furthermore, studies have shown that the exosomes isolated by coprecipitation may be contaminated by polymer molecules, including PEG [138,139,140].

In this sense, the differential expression of specific surface biomarkers, such as CD9, CD63, and CD83 provides a great opportunity for the development of immunoaffinity-capture-based techniques for exosomes isolation using modified magnetic beads or microchannels surfaces with specific antibodies [72]. Although this technique allows isolating only exosomes, it works with few volumes, limiting its use for therapeutic purposes, which require scalable methods. Moreover, this method generally requires a pre-enrichment step, which is commonly performed using commercial kits based on coprecipitation, which can result in PEG contamination.

Another strategy used to isolate exosomes is the size-based isolation technique. These methods allow isolating the EVs according to their size [72]. This technique can be based on sequential filtration, size-exclusion chromatography (SEC), and size-dependent microfluids.

In the sequential filtration, EVs are separated using membrane filters with different size or molecular weight exclusion limits. For this, firstly, the CM is filtered using a 0.22 μm membrane filter. Then, proteins with a 500 kDa molecular weight are filtered using a dialysis bag. Finally, the samples are filtered with a 100 nm membrane filter [72].

The SEC is based on particle size filtration through a porous stationary phase, which is composed of spherical gel beads that have pores of specific size [72]. When the sample passes through the stationary phase, large particles are eluted, whereas small particles pass through the pores [72]. The size-dependent microfluidics uses a viscoelastic microfluidics device composed of a microchannel, two inlets, and three inlets, to fractionate exosomes from other types of EVs [72]. All of these techniques are faster than those based on ultracentrifugation, and they do not require special equipment. Moreover, they avoid PEG contamination, which is frequently observed in coprecipitation-based methods. However, the size-based isolation techniques are relatively expensive and cannot separate exosomes from other EVs, requiring additional steps for exosome purification.

## 10. Stem Cell-Derived Exosomes as a Therapeutic Approach for Neurodegenerative Diseases

The blood–brain barrier severely limits the transport of large molecules to the brain. For this reason, it is estimated that 98% of all potent drugs that may improve the therapy of various diseases of the central nervous system (CNS) are not in clinic because of their inability to cross the blood–brain barrier (BBB) [141].

In this sense, the exosomes emerge as a promising therapeutic approach for neurodegenerative diseases [142,143]. This is because, due to their nano-sized diameter, the exosomes can cross the BBB, acting as carriers of bioactive molecules naturally secreted by their derived cells, or they can be biotechnologically engineered as carriers of drugs.

Parkinson’s disease (PD): In 2015, Haney et al. showed that exosomes can be permeabilized with saponins, which are a secondary metabolite derived from several vegetal species (Araldi), in order to load catalase, a potent antioxidant. Using in vivo models, the authors confirmed that the exosomes (administered by intranasal route) were taken up by neuronal cells, providing neuroprotective effects. Based on the therapeutic potential of catalase for PD, three years later, Kojima et al. [144] developed a series of synthetic biology-inspired control devices, which they call EXOsomal Transfer Into Cell (EXOtic) devices, to deliver the catalase mRNA delivery without the need to concentrate exosomes. These genetically encoded devices showed enhance exosome production, specific mRNA packaging, and delivery of the mRNA into the cytosol of target cells. Using this technology, the authors attenuated neurotoxicity and neuroinflammation in vitro and in vivo models of Parkinson’s disease, indicating the potential usefulness of the EXOtic devices for RNA delivery-based therapeutic applications [144]. A similar approach was also evaluated for delivering dopamine to the striatum and substantia nigra of PD mouse model, confirming that the exosomes can be used as a promising drug delivery platform for target therapy against PD and other neurodegenerative diseases [145]. In a current study, Chen et al. demonstrated that human umbilical cord mesenchymal stem cells (hucMSC)-derived exosomes can reach the substantia nigra through the BBB in vivo, reducing the dopaminergic neuron loss and apoptosis and upregulating the levels of dopamine in the striatum [146].

Alzheimer’s disease (AD): Cumulative evidence have been also demonstrated that the exosomes are a suitable therapeutic approach for AD as revisited by Guo et al. [147]. In a preclinical study, Reza-Zaldivar et al. [148] showed that MSC-derived exosomes promote neurogenesis and cognitive function recovery in a mouse model of AD. In another study, Ding et al. [149] demonstrated that the treatment with hucMSC-derived exosome in AD mouse models mimics the therapeutic effects of hucMSCs, alleviating neuroinflammation through the microglia activation, resulting in the repair of cognitive dysfunctions and helping clear Aβ deposition. Similar results were also reported by Elia et al. [150], who demonstrated that an intracerebral injection of extracellular vesicles from MSC exerts reduced Aβ plaque burden in early stages of a preclinical model of Alzheimer’s disease. The benefits of exosomal therapy were also reported by Nakano et al. [151], who recently showed that the treatment of AD mouse model with BM-MSC-derived exosomes increased the expression of microRNA-146a in the hippocampus, decreasing the levels of nuclear factor kappa B (NF-κB) in astrocytes, leading to synaptogenesis and the correction of cognitive impairment. Furthermore, MSC-derived exosomes are also shown to suppress the inducible nitric oxide synthase (iNOS) in cultured primary neurons and ameliorate the neural impairment of CA1 synaptic transmission in an AD mouse model [152]. In another recently published study, Cui et al. [153] developed brain-targeting exosomes derived from MSCs using CNS-specific rabies viral glycoprotein (RVG) peptide to increase the efficacy of intravenously delivered exosomes. Using these RVG-modified exosomes, the authors improved targeting to the cortex and hippocampus, reducing the expression of pro-inflammatory cytokines tumor necrosis factor alpha (TNF-α), interleukin 1 beta( IL-1β), and interleukin six (IL-6), improving the cognitive function in APP/PS1 mice [153].

Multiple sclerosis (MS): Studies have been also provided evidence that exosome-based therapy can be used as a treatment for multiple sclerosis. In a recent study, Clark et al. [154] showed that placental MSC-derived exosomes promote myelin regeneration in an animal model of multiple sclerosis. Similar results were described by Li et al. [155], who reported that MSC-derived exosomes reduced the infiltration of inflammatory cells into the CNS and decreased demyelination in the experimental autoimmune encephalomyelitis rat model (MS murine model). Similar results were observed with MSC-derived exosome intravenously administrated in Theiler’s murine encephalomyelitis virus (TMEV)-induced demyelinating disease (another model for MS) [156]. In this study, the authors showed that the EV administration attenuated motor deficits through immunomodulatory actions, diminishing brain atrophy and promoting remyelination.

Amyotrophic lateral sclerosis (LAS): In 2016, Bonafede et al. [157] showed that exosomes derived from the murine adipose-derived stromal cell are able to protect NSC-34 cells (which overexpress human SOD1(G93A) or SOD1(G37R) or SOD1(A4V) mutants) from oxidative damage, which is responsible for ALS-related damages. Similar results were also reported by Lee et al. [158], which showed that the treatment with adipose-derived stem cell normalized the phospho- cAMP response element-binding protein (CREB)/CREB ratio, modulating the mitochondrial dysfunction and SOD-1 aggregation.

## 11. Minimal Information to Study Exosomes

Based on the increasing number of publications describing the physiological and pathological functions of EVs, in 2014, the International Society for Extracellular Vesicles (ISEV) proposed the Minimal Information for Studies of Extracellular Vesicles (MISEV2014) guideline, which was updated in 2018 (MISEV2018) [11]. Based on the MISEV2018 and the last advances in the field of exosome characterization, here we bring minimal information to study exosomes.

For exosome isolation and characterizations from conditioned media, we endorse the recommendations of MISEV2018. Thus, we emphasize that the characterization of the releasing cells, culture, and harvest conditions must be performed and reported. In this process, we consider it mandatory to identify the cell lineage by short tandem repeat (STR) profiling especially for transcriptomic and proteomic analyzes of exosomes, since these data can be shared between laboratories through a web-based compendium such as ExoCarta [159]. In addition, it is crucial to determine the percent of dead cells at the time of exosome harvest, since an increased number of dead cells can lead to a large number of apoptotic bodies.

For therapeutic purposes, it is also important to standard the conditions for cell propagation, since it is known that the alteration in cell culture conditions, such as cell confluence, oxygen partial pressure (normoxia or hypoxia), culture medium composition, the concentration of growth factors, time of serum starvation, and even passage can alter the production/release of exosomes, as well as the exosome content.

Considering that the isolation method is critical to guarantee the physical integrity of exosomes [66], it is important to choose the appropriate isolation technique according to the purpose. In this sense, several strategies have been used to isolate and/or purify exosomes, as shown in Table 1. Despite the pros and cons of each isolation technique (previously discussed), the methods based on coprecipitation with PEG and immunoaffinity-capture have been successfully used for scientific purposes. However, for therapeutic purposes, the methods based on ultracentrifugation or filtration are most indicated, since these strategies can be scalable.

Once isolated, it is crucial to quantify and characterize the vesicles isolated, especially in relation to the size distribution [11]. In this sense, techniques providing images of single EVs at high resolution, such as electron microscopy (EM), atomic force microscopy (AFM), and super-resolution microscopy have been commonly used. Although these techniques provide important morphological parameters, they are not interchangeable in the information that they provide [11]. Furthermore, none of them allow quantifying the vesicles isolated.

In this context, single-particle analyses technique that estimates biophysical features of EVs, measuring the light scattering properties, such as nanoparticle tracking analysis (NTA), high-resolution flow cytometry, multi-angle light scattering coupled to asymmetric flow filed-flow fractionation or fluorescent properties allow determining the concentration of vesicles [11]. Among these methods, NTA is the most commonly used for analyzing the size distribution and the EV concentration [160,161,162].

NTA is a powerful characterization technique that combines the properties of both laser light scattering microscopy and Brownian motion in order to obtain size distributions of particles in liquid suspension [161,162]. NanoSight instruments (LM10 or NS300, Malvern, UK) are the most widely used instruments for NTA in the EV field [162]. These instruments are equipped with one or more lasers and an optical microscope connected to a digital camera. According to the manufacturer, NanoSight enables the characterization of particles from 10 to 2000 nm in solution [162]. EVs are visualized by the light they scatter upon laser illumination, and their Brownian motion is monitored. The NTA software enables the sizing of single particles by tracking their mean squared displacement and thereby calculating their theoretical hydrodynamic diameter using the Stokes–Einstein equation. On the basis of knowing the analyzed sample volume, NTA also allows for an estimation of particle concentration [162].

Analysis of the size distribution of homogeneous particle preparations of either the same or mixed sizes by NanoSight has previously been shown to be accurate, whereas analysis of heterogeneous biological vesicles has proven to be more challenging [162,163]. However, considering that the exosome size is closely related to the cell origin, as well as the culture conditions, the standardization of cell culture conditions can lead to homogenous particle preparations, making NTA a gold standard for determining the exosome concentration for both scientific and clinical purposes. However, the expertise of the operator is crucial to guarantee the results, since the operator adjustments of capture and analysis settings, such as camera level and detection threshold, have been shown to strongly affect the NanoSight-based quantification of EVs [163,164,165]. Moreover, it is known that NTA is most accurate between particle concentrations in the range of 2 × 10^8^ to 20 × 10^8^/mL. When samples contained a higher numbers of particles, they may be diluted before analysis, and then, the relative concentration is calculated according to the dilution factor [161].

Although flow cytometry (FC) has been widely used to characterize the EVs by the immunodetection of surface biomarkers (CD9, CD63, and CD81), this technique is not indicated for analyzing exosomes [166]. This is because the exosome size (30–0150 nm) is below the detection limit of FC (about 300 nm); this technique only detects a small fraction (1–2%) of the vesicles present in biological fluid samples, which are composed of microvesicles and/or apoptotic bodies [166].

Beyond the NTA, EV imaging plays an important role in revealing the spatiotemporal property of exosomes to further our understandings of molecular biology as well as the therapeutic potential of these vesicles [167]. In this sense, electron microscopy (EM) has been considered as a standard imaging method for characterizing the mechanisms of release and uptake of exosomes as well as their morphology [167,168,169,170,171].

Typically, EM has a resolution around 0.5 nm, which is smaller than exosomes. For this reason, this method has provided detailed structural information of EVs, including exosomes. However, EM cannot image EVs in their native state, once the samples need to be fixed and processed prior to imaging [167]. In addition, there are several EM methods used for EV imaging.

Transmission electron microscopy (TEM) is the most common type of electron microscopies for exosome imaging [172]. For TEM analysis, the exosomes are fixed and dehydrated. Following dehydration, the samples are embedded, sliced into nanometer-thin sections, and mounted on a carbon-coated grid for imaging. TEM uses electron beams to illuminate through prepared specimens, and the electron can either transmit or be diffracted by the specimens. A fluorescent screen or charge-couple device (CCD) will collect the transmitted electron for brightfield images, which is normally used for structure verification. Meanwhile, scattered electrons are collected to generate dark-field images, revealing the structure with higher contrast. EVs observed by TEM often appear as cup-shaped as a result of dehydration during sample preparation, but they can effectively reveal the inner structure of EVs. Using immunogold labeling, TEM can further reveal EV proteins [167].

Scanning electron microscopy (SEM) uses an electron beam to scan the surface of a specimen to generate topography information. For this analysis, samples are chemically or cryogenically fixed, which is followed by dehydration. Then, the immobilized samples are sputter-coated with a thin layer of a conductive material such as gold or carbon for imaging [167]. This thin layer of gold does not usually affect the imaging result. However, due to the small size of exosomes, the thin layer of gold may affect the surface structure of these vesicles [167]. However, a low-voltage SEM can avoid an accumulation of charge and reduce radiation damage to the samples, thus bypassing the sputter coating process [167].

Under SEM analysis, EVs can present round or saucer-shaped morphology [167]. However, the saucer-shaped morphology can be attributed to the EV collapse as a result of the dehydration process during sample preparation [167].

In cryo-electron microscopy (cryo-EM), samples are fixed by cryo-immobilization, where waters are vitrified instead of ice crystal formation in the sample by liquid ethane cooling [167]. Cryo-immobilizing allows samples to be preserved in their native hydrated state, thus avoiding artifacts commonly caused by conventional fixation methods such as cup-shaped EVs [167].

Under cryo-EM, the specimens are imaged under extremely low temperatures (below—175 °C) as EVs are maintained in its original spherical shape [167]. Therefore, the average size of EVs will appear to be bigger when compared to other EM methods [167,173]. Since cryo-EM yields superior sample quality and morphology preservation over traditional EM methods, it is increasingly being applied to study EVs [167,173].

However, due to the resolution limit of the EM, novel imaging strategies have been used for analyzing the exosomes. Among these techniques, atomic force microscopy (AFM) has been considered the most accurate for imaging exosomes, since this method generates topographic images with a resolution limit around 1 nm [166,167].

AFM is a versatile scanning probe technique widely used to image and study nano-objects and nanomaterials with a resolution down to the nanoscale [166]. The ability of imaging soft samples without damaging in different environmental media (air or liquid) makes AFM a powerful tool to study biological samples, including exosomes under physiological conditions [166]. For this analysis, the EVs are immobilized on a selective antibody-modified mica substrate or on a poly-L-lysine-modified mica substrate and characterized by tapping-mode AFM [166]. The morphology and size of the EVs are compared under liquid and dry conditions [166]. This is because according to Sebaihi et al. [166], the EV measurements in air are 6–10 times smaller than those in liquid.

Thus, considering the technical aspects of each method, we recommend that the exosome characterization may be performed using ETM, AFT, and NTA.

In past years, analyses of proteomic and transcriptomic content of EVs have been added to the characterization process of these vesicles.

Basically, for the proteomic studies, two approaches for characterizing the exosome content have been used: shot-gun and immunological assays [44].

If on hand the immunological assays, which include enzyme-linked immunosorbent assay (ELISA), Luminex antibody bead-based array, cytokine antibody array, and Western blot, offer high specificity, sensitivity, and reproducibility, on the other hand, they only allow identifying known proteins [44]. For this reason, the shotgun-based proteomic approach has been commonly used, since this method is more exploratory, allowing the identification of any protein [44]. This is because the protein identity can be determined by accessing a publicly available database using bioinformatics tools [44]. In this sense, different techniques have been successfully used to evaluate the proteomic profile of the MSC secretome, including exosomes, 2D gel electrophoresis (2-DE), liquid chromatography with tandem mass spectrometry (LC-MS/MS), stable isotope labeling by amino acids in cell culture (SILAC), matrix-assisted laser desorption/ionization–time of flight (MALDI-TOF), MS/MS and quadrupole time-of-flight mass spectrometry (QTOF-MS) [44,174,175].

Another strategy for identifying proteins present in exosomes is the proteomic profiler array. Although this method allows identifying more than 20 proteins simultaneously (according to the array used), it is less sensitive than mass spectroscopy analysis. However, the proteomic profiler array emerges as a powerful technique to investigate the phosphorylation levels of different proteins as well as to study proteins from a single pathway. In this sense, different arrays are commercially available (https://www.rndsystems.com/products/proteome-profiler-antibody-arrays).

Considering that both mRNA and non-coding RNA can regulate cell physiology at a post-transcriptional level, it is also recommended to analyze the transcripts present in the exosomes [176]. For this, firstly, it is necessary to isolate the RNA from the EVs. Although there are several commercial kits for RNA isolating, Prendegast et al. [177] showed that the Trizol^®^ reagent optimizes the RNA isolation for RNA sequencing (RNA-seq).

RNA-seq was developed more than a decade ago, and since then, it has become a powerful tool in molecular biology, allowing understanding the genomic function [178]. RNA-seq is most often used for analyzing differential gene expression (DGE), allowing comparison of the transcriptome of parental cells with their respective exosomes [178].

The standard workflow of RNA-seq begins in the laboratory, with RNA extraction, followed by mRNA enrichment or ribosomal RNA depletion, cDNA synthesis, and the preparation of an adaptor-ligated sequencing library. Then, the library is sequenced to a read depth of 10–30 million reads per sample on a high-throughput platform (usually Illumina). The final steps are computational: aligning and/or assembling the sequencing reads to a transcriptome, quantifying reads that overlap transcripts, filtering and normalizing between samples, and statistical modeling of significant changes in the expression levels of individual genes and/or transcripts between sample groups [178].

Thus, we recommend the combined use of the approaches discussed in this section to confirm, characterize, and validate the exosomes purified for both scientific and clinical purposes.

## 12. Perspectives

For the first time in history, most people can expect to live into their sixties and beyond. According to the World Health Organization, by 2050, the world’s population aged 60 years and older is expected to total 2 billion (https://www.who.int/news-room/fact-sheets/detail/ageing-and-health). As a consequence of the population aging, we are observing a rapidly rising in the global incidence of neurodegenerative disorders. Despite the medical advances, there are no effective treatments for neurodegenerative disorders, and the treatments available only manage the symptoms or halt the progression of the disease. Therefore, there is an urgent need for new treatments for these diseases, since the World Health Organization has predicted that neurodegenerative diseases affecting motor function will become the second-most prevalent cause of death in the next 20 years [179]. In this sense, cell therapy, using stem cells, has been recognized as the best candidate for treating incurable diseases, including neurodegenerative disorders. However, in the last decade, accumulating evidence supports the idea that MSCs perform their therapeutic roles in a paracrine manner [180]. This idea has been led a new way of exploring the therapeutic potential of stem cells without the need for cell transplantation [46]. In this sense, the rediscovery that cells secrete a plethora of factors into nanosized vesicles surrounded by a lipid bilayer membrane (extracellular vesicles), which confer thermostability to these factors, has allowed exploring the therapeutic use of these vesicles in a novel therapeutic modality known as cell-free therapy. Despite the evidences of the benefits of cell-free therapy for neurodegenerative diseases, efforts are necessary to improve the available exosome isolation in order to realize scalable vesicles production for clinical purposes. In addition, it is also crucial to provide guidelines for studying these vesicles in order to guarantee acceptance criteria by regulatory agencies.

## Figures and Tables

**Figure 1 cells-09-02663-f001:**
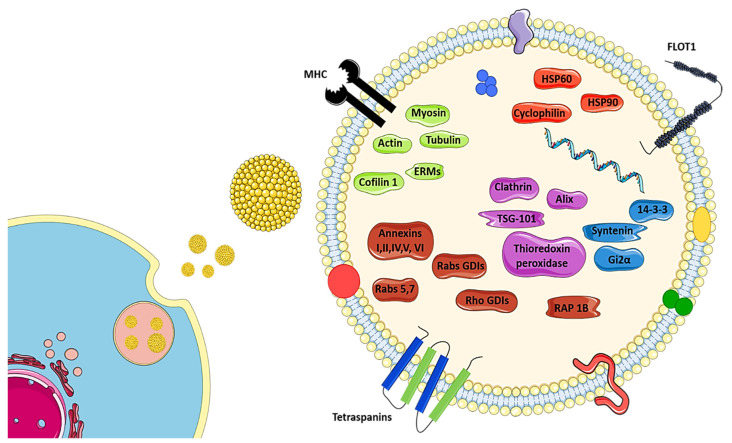
Illustration of exosome release, showing the main exosome biomarkers described. Figure adapted from Kourembanas et al. [92].

**Figure 2 cells-09-02663-f002:**
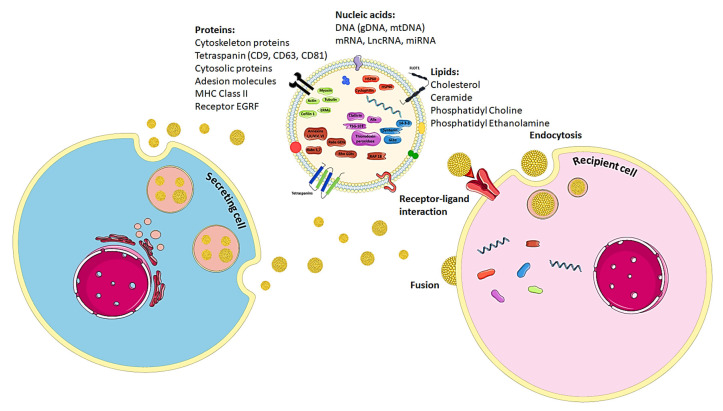
Illustration of exosome and uptake by the recipient cell. Exosomes released by the secreting cell can be internalized by both secreting and recipient cells by endocytosis through the interaction between the exosome with the cell membrane or through the interaction with specific receptors, leading to an autocrine or paracrine signaling.

**Table 1 cells-09-02663-t001:** Methods commonly used to isolate extracellular vesicles.

Noncommercial Methods			
Methods	Procedures	Time	Reference
**UC**	Clearance: CM is collected and centrifuged at 2700× *g* for 5 min and afterwards filtered using a 0.2 μm filter. Isolation: Supernatant is centrifuged at 100,000× *g* overnight at 4 °C	>16 h	Klymiuk et al. [49]
**UC**	Clearance: CM is collected and filtered using 0.2 μm filter. Filtrate is subjected to two centrifugation steps: (1) 2000× *g* for 10 min and (2) 10,000× *g* for 70 min. Isolation: Supernatant is centrifuged at 100,000× *g* for 70 min and afterwards, washed in PBS and centrifuged at 100,000× *g* for 70 min	>3 h	Klymiuk et al. [49]
**UC**	Clearance: CM is collected and subjected to two centrifugation steps: (1) 300× *g* for 10 min and (2) 10,000× *g* for 30 min. Next, the supernatant is filtered using a 0.2 μm filter. Isolation: Filtrate is centrifuged at 100,000× *g* for 90 min, discarding the supernatant. Exosome pellet is washed in PBS and then centrifuged 100,000× *g* for 90 min	≈3 h	Haraszti et al. [67]
**UC**	Clearance: CM is collected and subjected to two centrifugation steps: (1) 300× *g* and (2) 1200× *g*, both for 10 min. Supernatant is filtered using 0.2 μm filter. Isolation: Filtrate is centrifuged at 100,000× *g* for 14 h at 4 °C.	>14 h	Antounians et al. [89]
**UC**	Clearance: CM is collected and subjected to two centrifugation steps: (1) 400× *g* for 10 min and (2) 2000× *g* for 20 min. Supernatant is filtrated using a 0.2 μm filter. Isolation: Filtrate is centrifuged at 100,000 × *g* for 3 h	≈4 h	Duong et al. [134]
**UC**	Clearance: CM is collected and subjected to two centrifugation steps: (1) 300× *g* and (2) 1200× *g*, both for 10 min. Isolation: Filtrate is centrifuged at 100,000× *g* for 14 h	≈15 h	Antounias et al. [89]
**UC**	Clearance: CM is collected and centrifuged at 2700× *g* for 5 min and afterwards filtered using a 0.2 μm filter. Filtrate is subjected to two centrifugation steps: (1) 2000× *g* for 10 min and (2) 10,000× *g* for 30 min. Isolation: Supernatant is centrifuged at 100,000× *g* for 70 min, discarding the supernatant. Exosome pellet is washed in PBS and then centrifuged 100,000× *g* for 70 min.	≈4 h	Klymiuk et al. [49]
**UC**	Clearance: CM is collected and subjected to three centrifugation steps: (1) 300× *g* for 10 min, (2) 2000× *g* for 30 min, and (3) 20,000 for 30 min. Isolation: Supernatant is centrifuged at 100,000× *g* for 70 min, discarding the supernatant. Exosome pellet is washed in PBS and then centrifuged 100,000× *g* for 70 min.	≈4 h	Narbute et al. [135]
**CUC**	Clearance: CM is collected and subjected to two centrifugation steps: (1) 400× *g* for 10 min and (2) 2000× *g* for 20 min. Supernatant iscollected Isolation: Filtrate is loaded on 60% iodixanol cushion. Sample is centrifuges at 100,000× *g* for 3 h.	≈4 h	Duong et al. [134]
**CUC**	Clearance: CM is collected and subjected to two centrifugation steps: (1) 300× *g* and (2) 6000× *g*, both for 30 min. Supernatant is filtrated using a using a 0.2 μm filter. Isolation: Filtrate is loaded on 30% sucrose/D_2_O cushion. Samples is centrifuged at 120,000× *g* for 90 min, discarding the supernatant. Exosome pellet is washed in PBS and next, centrifuged at 120,000× *g* for 90 min.	≈6 h	Li et al. [136]
**DGUC**	Clearance: CM is collected and subjected to two centrifugation steps: (1) 400× *g* for 10 min and (2) 2000× *g* for 20 min. Supernatant is filtrated using a 0.2 μm filter. Isolation: Exosomes, previously isolated using UC method, is loaded on 5, 10 or 20% iodixanol gradient solution with 0.25 mm sucrose, 1 mM Tris-HCl pH 7.4 and centrifuged at 100,000× *g* for 18 h	≈20 h	Duong et al. [134]
**PEG-B**	Clearance: not applied Isolation: CM is collected and transferred to centrifugation tubes containing 1:4 (*v*/*v*) of 50% PEG 6000 with 375 mm NaCl—CM. Sample is incubated for 18 h at 4 °C and next, centrifuged at 1500× *g* for 30 min.	≈19 h	Duong et al. [134]
**PEG-B**	Clearance: CM is collected and centrifuged at 3000× *g* for 20 min. Supernatant is filtrated using a 0.2 μm filter. Isolation: Filtrate is incubated with 50% PEG 35.000 and 2% protaminesulfate for 1 h at 4 °C and, next, centrifuged at 12,000× *g* for 10 min.	≈2 h	Klymiuk et al. [49]
**UF**	Clearance: CM is collected and centrifuged at 3000× *g* for 20 min. Supernatant is filtrated using a 0.2 μm filter. Isolation: Filtrate obtained is filtrated using 3K Ultra Filter and then, centrifuged at 2700× *g* for 10 min. Pellet is washed with PBS and, next, centrifuged at 2700× *g* for 10 min.	≈0.5 h	Klymiuk et al. [49]
**TFF**	Isolation: CM is subjected to ultrafiltration in a tangential flow filtration (TFF) system, using a 500 kDa cutoff TFF cartridge with a flow rate of 120 mL/min and a transmembrane pressure < 3.5 psi. Then, the filtrate is concentrated 9-fold and exchange with 6× volume of PBS. Finally, the exosomes are filtrated using a 0.2 μm filter.	≈2 h	Haraszti et al. [67]
**Commercial Methods**			
**Methods**	**Procedures**	**Time**	**Reference**
**Total Exosome Isolation (TEI)**	Procedure: CM is collected and centrifuged at 2000× *g* for 30 min at 4 °C. Supernatant is incubated overnight at 4 °C with the Total Exosome Isolation reagent. Then, the sample is centrifuged at 10,000× *g* for 60 min at 4 °C.	>20 h	Thermo Fischer Scientific
**ExoQuick Preciptation Solution**	Procedure: CM is collected and centrifuged at 3000× *g* for 15 min at 4 °C. Supernatant is filtered using a 0.2 μm filter. Filtrated is incubated overnight at 4 °C with the ExoQuick reagent. Then, the sample is centrifuged at 1500× *g* for 5 min at 4 °C.	>20 h	System Biosciences
**ExoMAX Opti Enhancer Reagent**	Procedure: CM is collected and centrifuged at 3000× *g* for 30 min (clearance). Supernatant is incubated overnight with ExoMAX Opti Enhancer Reagent. Then, the sample is centrifuged at 1500× *g* for 30 min. Pellet is washed and centrifuged at 75,000× *g* for 70 min.	>20 h	BioCat
**qEV Exosome Isolation**	Procedure: CM is collected and centrifuged at 16,000× *g* for 30 min at 4 °C (clearance). Supernatant is incubated overnight at 4 °C with the Exo-Spin reagent. Then, the sample is centrifuged at 16,000× *g* for 60 min at 4 °C.	>20 h	Cell Guidance Systems
**MagCapture^TM^ Exosome Isolation Kit**	Procedure: CM is collected and loaded on a stationary phase consisting of porous resin particles, in which particles between 35–350 nm enter into the pores.	≈2 h	IZON
**Exo-Spin^TM^ Exosome Purification**	Procedure: CM is collected and centrifuged at 300× *g* for 10 min. Next, the supernatant is filtered using a 0.2 μm filter (clearance). Filtrate is incubated overnight with Tim4 protein solidified magnetic beads able to bind to phosphatidylserine on the surface of extracellular vesicles. Finally, exosomes are captured using a magnetic rack.	>20 h	Wako

UC—ultracentrifugation; CUC—cushion ultracentrifugation; DGUC—density-gradient ultracentrifugation, PEG-B—polyethylene glycol-based method, UF—ultrafiltration, TFF—tangential flow filtration.

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
