# Peer review of "Stem Cell-Derived Exosomes as Therapeutic Approach for Neurodegenerative Disorders: From Biology to Biotechnology"

_cells, 2020, doi:10.3390/cells9122663_

Round 1
Reviewer 1 Report
The authors comprehensively reviewed the relations of exosome with neurodegenerative diseases in various aspects including disease mechanism, cell therapy effect and its own application as free-cell therapy. They also overviewed the detail history, nature, extraction and measurement methods of exosomes. By the way, it should be more attractive to readers if they add a little more specific knowledge and/or idea about the application of exosomes to neurodegenerative diseases.
Author Response
Point 1. By the way, it should be more attractive to readers if they add a little more specific knowledge and/or idea about the application of exosomes to neurodegenerative diseases
We thanks the suggestion and, aiming to add more details about the application of exosomes as a therapeutic approach for different neurodegenerative diseases, we include a new section (section 10) - lines 547-607
Reviewer 2 Report
Rodrigo Pinheiro Araldi and co-authors give an overview of the nature of MSC derived exosomes and their possible use in neurogenerative disease treatment. They also include a dissertation on extracellular vesicles and how to isolate and characterize them. Even if the review appears well done overall, there are some points that should be addressed:
The part regarding therapeutic uses of EVs may be expanded. The review seems to focus more on the general aspects of EVs than on their particular clinic use.
When talking of precipitating polymers, the lower purity of samples that can be obtained should be underlined.
Minor points:
In my opinion the title doesn’t appear to be fully pertinent since the entire review focuses on a particular field (EVs) rather than a multitude of cell-free techniques. This focus should be somehow present in the title.
Even though EVs are universally recognized as a promising therapeutic tool, it is clear that their production, quantitation and use are at least hard to standardize. The authors mention this fact in the review. From personal experience I can say that a way to bypass the problem might be inducing the production of in vivo engineered exosomes, or modified endogenous exosomes. the authors may want to check literature for in vitro and in vivo studies.
A grammar check is needed, for instance lines 52, 87, 202, 273, 398, 443.
Author Response
Point 1. The part regarding therapeutic uses of EVs may be expanded
We thanks for this suggestion and, aiming to provide more details exploring the therapeutic potential of exosomes for the treatment of neurodegenerative disease, we included a new section (section 10) - lines 550-610
Point 2. When talking of precipitating polymers, the lower purity of samples that can be obtained should be underlined
The information about the low purity of exosomes isolated by polymer-based coprecipitation was added in lines 524-525
Point 3. In my opinion the title doesn’t appear to be fully pertinent since the entire review focuses on a particular field (EVs) rather than a multitude of cell-free techniques. This focus should be somehow present in the title
Based on this comment, we changed the review title to "Stem cell-derived exosomes as therapeutic approach for neurodegenerative disorders: From biology to biotechnology" in order to focus in therapeutic action of exosomes.
Point 4. Even though EVs are universally recognized as a promising therapeutic tool, it is clear that their production, quantitation and use are at least hard to standardize. The authors mention this fact in the review. From personal experience I can say that a way to bypass the problem might be inducing the production of in vivo engineered exosomes, or modified endogenous exosomes. the authors may want to check literature for in vitro and in vivo studies.
These informations were included into de section 10 (lines 547-607), focus on in vivo studies
Point 5. A grammar check is needed, for instance lines 52, 87, 202, 273, 398, 443
We apologize for these grammar mistakes and inform that they were corrected